# Could Having Access to Real-Time Data on Your Emotions Influence Subsequent Behavior? Evidence from a Randomized Controlled Trial of Japanese Office Workers

**DOI:** 10.3390/bs14030169

**Published:** 2024-02-23

**Authors:** Yoshihiko Kadoya, Sayaka Fukuda, Mostafa Saidur Rahim Khan

**Affiliations:** School of Economics, Hiroshima University, 1-2-1 Kagamiyama, Higashihiroshima 7398525, Japan; m232090@hiroshima-u.ac.jp (S.F.); khan@hiroshima-u.ac.jp (M.S.R.K.)

**Keywords:** behavioral modifications, emotional status, mental health, real-time feedback, wearable biometric devise

## Abstract

Improvements in mental health through real-time feedback on emotions have consequences for productivity and employee wellness. However, we find few extant studies on how real-time feedback on emotions can influence subsequent behavior modification in the Japanese workplace. We conducted a randomized controlled trial (RCT) with 30 employees of an insurance company in Japan and observed their emotions for 10 working days using a wearable biometric device. We compared the emotions of employees who had access to real-time emotional states (treatment group) with those of employees who did not (control group). The results of the panel regression analysis showed that access to real-time emotions was negatively associated with happy emotions and positively associated with angry and sad emotions. The results indicated that even after having access to the objective statuses of emotions, participants were unable to continue with happy emotions and reverse angry and sad emotions to other comfortable emotions. Our findings imply that feedback on real-time emotional states should be associated with appropriate training and motivation to utilize feedback for behavioral modification.

## 1. Introduction

Improving mental health in workers is a key factor not only for their personal well-being but also for overall productivity and workplace safety [1,2,3]. The substantial costs associated with reduced mental health conditions and subsequent productivity loss have led management to prioritize strategies such as training initiatives, enhanced employee benefits, regular health assessments, inclusive coordination, equity promotion, and the implementation of positive reinforcement to support and sustain employee mental well-being [4]. Among these strategies, ensuring the sustainability of autonomous self-regulation through a real-time understanding of objective emotional states emerges as a potential solution to this ongoing challenge. Real-time feedback on emotional states has the potential to influence employee behavior in a direction conducive to desired outcomes. Although behavioral modification is essential in various cognitive and psychological theories [5,6,7,8,9], its application in altering employee emotions, particularly through real-time measurement of objective emotional states, remains relatively unexplored. Addressing this gap, our study seeks to test the hypothesis that real-time feedback on emotional states can effectively modify subsequent behavior. To examine this hypothesis, we conducted a panel study utilizing biomedical emotion tracking devices and used the Russel circumplex model to offer real-time measurements of emotional states [10]. The premise is that feedback regarding emotions helps people evaluate their actions, make adaptive choices, learn from mistakes, navigate social scenarios, and regulate their behavior as needed [11,12,13]. Integrating emotional feedback could potentially enable individuals to modify subsequent behavior, thereby achieving desired outcomes and enhancing overall well-being [11,12,13].

Behavioral modifications through feedback on emotional status find grounding in numerous behavioral and psychological theories such as operant conditioning, reinforcement theory, cognitive–behavioral theory, and self-regulation models [5,14,15,16,17,18,19,20]. This feedback mechanism capitalizes on the notion that behavior is shaped by its consequences [5,7,9]. By providing individuals with immediate information about their emotional responses to their actions, it establishes a direct link between behavior and emotions. Positive emotions function as reinforcement for desired behaviors, while negative emotions discourage unwanted actions, facilitating behavioral modification [5,7,9]. Moreover, real-time emotional feedback aligns with cognitive–behavioral theory by helping people identify and challenge maladaptive thought patterns and behaviors, promoting healthier alternatives [6,7]. Additionally, it empowers individuals to self-regulate their actions, fostering autonomy and control over their behavior [21]. This feedback loop, ingrained in human learning and adaptation processes, continually guides individuals toward favorable emotional outcomes and improved behavioral patterns.

Existing empirical studies have explored how feedback systems can redirect emotions toward desired states, often utilizing reinforcement and modification [11,12]. Emotions serve multiple roles in this context: reinforcing or discouraging specific behaviors, guiding decision making by signaling potential dangers or favorable outcomes, and serving as internal self-regulation feedback signals [13]. Individuals tend to repeat actions that evoke pleasure or delight when praised, whereas negative emotions such as remorse or shame deter such behaviors [22,23]. Furthermore, emotions play a pivotal role in decision making, influencing individuals to avoid certain activities driven by fear, anger, or sadness while motivating actions through positive emotions such as joy or excitement [22,23,24]. Moreover, emotions serve as signals of self-regulation, urging individuals to reflect on and potentially modify their actions to achieve better results [24].

The primary objective of this study is to explore whether real-time feedback on emotional states can effectively steer subsequent behavior in the desired direction. Behavioral modifications play a key role in productivity and health, with positive behaviors that promote productivity while negative behaviors hinder it [1,2,3]. Although theories such as operant conditioning, reinforcement theory, cognitive–behavioral theory, and self-regulation models propose that behavioral modification is achievable through reinforcement and deterrent mechanisms [5,14,15,16,17,18], empirical research exploring behavioral modifications after feedback on immediate past behavior remains limited. This study seeks to bridge this gap by investigating the correlation between real-time emotional feedback and deliberate changes in subsequent behavior. Our contribution lies in providing empirical evidence on how emotional feedback can be utilized to modify subsequent behavior, addressing critical gaps in the literature.

To investigate how real-time feedback on emotions influences subsequent behavior, we conducted a randomized controlled trial (RCT) involving 30 office workers from the same department within a Tokyo-based insurance company. These participants wore biometric emotion tracking devices for 10 working days, categorizing emotions into five states (happy, angry, relaxed, sad, and neutral) based on the Russel circumplex model [10]. Subsequently, we randomly assigned them into two groups: a treatment group with access to their objective emotional status via a smartphone and a control group without such access. This design enables us to observe whether objective emotional feedback impacts the behaviors and mental health of workers.

The rest of this study is organized as follows: Section 2 provides a literature review, Section 3 outlines the data and methodology, Section 4 presents empirical findings, Section 5 discusses the results, and Section 6 concludes this paper.

## 2. Literature Review

The impact of workers’ mental health on productivity has been increasingly recognized in academic research [1]. Studies such as Kadoya et al.’s investigation into factory workers’ emotional status, tracked through biometric devices, revealed a direct correlation with their productive output [1]. Similarly, Kadoya et al. found a connection between tracked emotional states and safe driving behaviors among professional taxi drivers [2]. Agnafors et al. also contributed to this field, emphasizing the detrimental effect of poor mental health on academic performance [3]. However, these studies lack an intermediate path to understand the association between mental well-being and productivity, and whether behavioral modifications can alter emotions [25,26,27,28].

This gap has fueled an exploration into fostering sustainable mental health solutions, particularly through autonomous self-regulation. One proposed approach involves making objective emotional states, tracked by biometric devices, accessible to workers [1]. Long-term stress is often associated with poor mental health, but self-awareness of one’s mental state can empower individuals to recognize and deal with stressors effectively [4,5,6].

Behavioral modifications and emotional regulation are critical methods in achieving desired emotional states [7]. Previous research has underscored the importance of emotional regulation, demonstrating its influence on subsequent cognitive and behavioral changes [8,9]. Positive emotions incentivize preferred behaviors, while negative emotions discourage undesired actions, aiding in behavioral adjustments [5,7,9]. However, the effectiveness of emotional modification depends on context and often requires training, medications, or stimulation [29,30,31].

The continuous evaluation of emotion through biometric devices, coupled with feedback provision, can guide individuals in redirecting their emotions to desired states [4,5,6]. Reinforcing positive emotions through behavioral modification manifests itself in multiple ways. Firstly, emotions can reinforce or discourage specific tasks, influencing future behavior based on the experienced emotional outcomes [7,8]. Secondly, emotions significantly impact decision making, with fear, anger, and sadness deterring certain actions while joy or excitement motivate favorable behaviors [7,8,22]. Thirdly, emotions serve as self-regulation feedback signals; negative emotions prompt reflection and potential behavior changes for better future outcomes [22].

Although mental health measurement scales are crucial in evaluating intervention effectiveness, the utilization of biometric devices for real-time feedback, as demonstrated by Kadoya et al., remains a unique and promising approach [1,2]. Studies in various fields, such as driving, confirm the efficacy of biometric devices in objectively detecting emotions and their association with performance and behavior [32,33,34,35,36,37,38]. However, applying emotional feedback as a tool for behavioral modification within employee settings remains an understudied area that warrants further investigation.

This study aims to address critical gaps in the literature on emotion and subsequent behavioral modification. Primarily, the deficiency lies in the absence of research using real-time emotion, complicating conclusive insights on feedback efficacy for altering subsequent behavior. Furthermore, empirical studies elucidating the process by which feedback influences behavioral change are notably lacking. To bridge these gaps, our research employed a biometric device to measure real-time emotions, providing participants with immediate feedback without additional behavioral modification support. This approach allowed us to examine whether feedback alone is adequate to guide subsequent behavior toward the desired outcomes. We hypothesized that human behavior and cognitive skills play a pivotal role in modifying emotions, aligning with theories such as operant conditioning, reinforcement theory, cognitive–behavioral theory, and self-regulation models [5,6,14,15,16,17,18]. People utilize reinforcement and deterrent mechanisms to decide which emotions to perpetuate and which to modify. Therefore, we anticipated that individuals exposed to positive emotional feedback would maintain happier and relaxed emotional states, while those receiving negative feedback would tend to transition toward more positive emotional states.

## 3. Data and Methodology

We adhered to the standard procedure for conducting a randomized control trial in accordance with CONSORT 10 guidelines, as detailed in relevant studies [39,40].

### 3.1. Participants

This study was conducted at Aioi Nissay Dowa Insurance Co., Ltd. (Tokyo, Japan), a renowned Japanese insurance company established in 1918. The participant selection criteria mandated individuals to be 20 years or older, employed as full-time staff members of the company, and to possess a minimum of two years of experience. The only exclusion criterion was serious illness, leading to an absence from the workplace for a week or more among employees. We randomly contacted employees in the same department to inform them about the purpose and investigation process, and they voluntarily agreed to participate. We applied Cohen’s methodology to determine the minimum sample size. With an alpha value of 0.05, a power level of 0.8, and an effect size of 0.5, the calculated minimum sample size was 30.47. This study’s sample consisted of 30 employees aged 23–59 years, each with a tenure at the company ranging from 2 to 33 years. There were 17 male and 13 female participants. The average age of the participants was 37.83 years, with an average of 12.37 years of work experience. Their emotional state and working situation (such as working hours and whether they took sick leave) were observed for 10 working days (two weeks, excluding two weekends) from 14 to 28 October 2022. We excluded the data collected on the first day of the experiment as the participants attended an experiment explanation session, provided their consent forms, and completed a socioeconomic-related questionnaire. Before conducting the investigation, the participants were informed about the purpose, methods, and privacy issues related to this study. All participants agreed to participate in this study and signed an informed consent form. Participants were also informed that they could withdraw from this study at any time. The ethics committee of Hiroshima University, Japan, approved the study protocol (approval number: E-2826).

### 3.2. Experimental Design

The information that the participants provided through the questionnaire on the first and last days after work hours included demographic and socioeconomic information such as gender, date of birth, marital status, education, family background, and subjective health information such as loneliness. During the experiments, the participants also answered a daily-based questionnaire about their work experience that day, including working hours, whether they were home- or office-based, and lunch breaks.

To track and record the emotional states of the participants, we asked them to wear a wristband biometric device that captured their physiological responses for the duration of the experiment. The Silmee^TM^W22 device was manufactured by TDK Corporation in Tokyo, Japan. The device measures 52 × 24.5 × 13.5 mm and weighs approximately 26 g. It contains built-in sensors that detect acceleration, pulse waves, environmental ultraviolet light, temperature, and sound. This enables it to continuously record physical activity, beat-to-beat pulse intervals, skin temperature, and sleep. The device also measures conversation time (i.e., speaking during a specific time interval), the number of steps per minute, and the type of activity (not moving, walking, running, light–medium–heavy exercise, and sleep state). Using the software developed by NEC Corporation in Tokyo, Japan, this device was also used in recent studies by Miyamoto et al. [41] and Hayano et al. [42] to measure emotions (happy, relaxed, sad, and angry), and Kadoya et al. used it to study factory workers and taxi drivers [1,2].

The measurement of emotional conditions followed Russell’s [10] circumplex model of emotion, which indicates that emotions are distributed along the dimensions of arousal and valence. Rubin and Talarico [43] showed that arousal represents the vertical axis, valence represents the horizontal axis, and the center represents a neutral valence. According to this model, emotional states represent different levels of activity and pleasantness (or unpleasantness). In this study, we focused on happy, angry, relaxed, and sad emotional states. We considered it neutral in the absence of distinct emotions. With the biometric device, emotional states were measured through a complex process. In the measurement process, the beat-to-beat pulse intervals were processed to detect the periods of noise and data defects via custom software developed by NEC Corporation. In addition, the condition of each subject’s automatic nervous system was evaluated by observing the R-R interval (RRI), which indicates heartbeat variability. The software uses a specific algorithmic pattern of the subject’s heartbeat variability to differentiate between emotional states. TDK Corporation processes all the information recorded by the biometric device and classifies the emotional states of the drivers as happy, angry, relaxed, sad, and neutral. Hayano et al. [42] adopted a similar procedure to measure emotional states and found that the pulse rate increased in the order of angry, happy, neutral, relaxed, and sad emotions.

We employed a simple randomization process to allocate participants into the control and treatment groups. Initially, we assigned random numbers to employee seats to determine their allocation into either group. Subsequently, we used a coin flip to finalize the assignment of each employee to either the treatment or control group. Only the treatment group received a URL with which to access their own objectively measured emotional status. Members of the control group were unaware of the specific URL, and more importantly, they remained unaware that the treatment group had access to it. Finally, to test our hypothesis, we treated the emotional state as the dependent variable and the dummy variable for the treatment group as the main independent variable. Moreover, we controlled for several variables related to participants’ demographic and socioeconomic backgrounds to isolate the effect of feedback. The definitions of all the variables used in this study are listed in Table 1.

### 3.3. Descriptive Statistics

The dependent variables of this study were emotional states (neutral, happy, angry, relaxed, sad), and the main independent variable was the dummy variable for the treatment group, who were given access to observe their real-time emotional states. Descriptive statistics of the main variables are shown in Table 2. The entire sample comprises 300 observations based on the data of 30 employees collected over 10 working days. However, some data on the variables of emotional and working status were missing due to technical problems with the biometric device (the devices did not capture the biometric information of the participants when the device was not properly worn) and the work schedule of the participants (some participants took sick leave during the experiment). Kadoya et al. [2], whose experiments utilized the same experimental setting as in the current study, also reported that the device sensors sometimes cannot capture data when they lose their connection to the skin. Despite these limitations, we collected more than 90% of the data. The number of observations for this study was 277.

According to Table 2, in general, the subjects remained in happy, angry, relaxed, and sad states at 28%, 36%, 10%, and 4%, respectively. An average of 43% of employees engaged in remote work. Moreover, they worked for an average of 630 min (SD = 97), commuted for 54 min (SD = 19) per day, and took a break by walking or exercise for an average of 4% of their working hours. Regarding socioeconomic variables, 17 employees were male and the average age was 38 years (SD = 10). The percentages of married employees, having a bachelor’s degree, living alone, and having children of preschool age were 57%, 87%, 37%, and 17%, respectively. The average household assets of the employees were 12,900,000 JPY (SD = 18,600,000) and 16.67% of employees exercised regularly. Regarding psychological variables, the percentages of employees with health anxiety, loneliness, and myopic views were 37%, 60%, and 47%, respectively. Finally, participants used smartphones for an average of 264 min (SD = 208).

### 3.4. Metholody

This study used a panel data analysis technique because the data of 30 employees were collected over 10 working days. The Breusch–Pagan Lagrange multiplier (LM) test was used to determine whether the random-effects model or pooled ordinary least-squares model was appropriate. The null hypothesis of the LM test, variance across entities, is zero (i.e., there is no significant difference across units), and if the null hypothesis is rejected, the random-effects model is appropriate. The results of the LM test (available upon request) indicate that the null hypothesis was rejected in almost all models. Following the results of the LM test, we adopted a random-effects model. We used estimation equations for each emotional state to gain a deeper understanding on the influence of work-related and other variables. For the estimation equations in the first specification (Equation (1)), only work-related variables were included in the model to explain emotional status. In the second specification (Equation (2)), besides daily working variables, employees’ personal characteristics were added to control for demographic factors. Personal-characteristic variables included age, marital status, university degree, household assets, duration of smartphone use, and psychological status. The regression equations were as follows:(1)$$\begin{array}{ll} {Emotional\;status}_{it} & \left(neutral,\;happy,\;angry,\;relaxed,\;and\;sad\right)\\ & =\beta_{0}+\beta_{1}{treatment}_{it}+\beta_{2}{worktime}_{it}+\beta_{3}{break\_walk\_exercise}_{it}+u_{i}+\varepsilon_{it} \end{array}$$
(2)$$\begin{array}{ll} {Emotional\;status}_{it} & \left(neutral,\;happy,\;angry,\;relaxed,\;andsad\right)\\ & =\beta_{0}+\beta_{1}{treatment}_{it}+\beta_{2}{worktime}_{it}+\beta_{3}{break\_walk\_exercise}_{it}+\beta_{4}{male}_{it}+\beta_{5}{age}_{it}\\ & +\beta_{6}{married}_{it}+\beta_{7}{university\_degree}_{it}+\beta_{8}{living\_alone}_{it}+\beta_{9}{children}_{it}+\beta_{10}{travel\_time}_{it}\\ & +\beta_{11}{log\_hasset}_{it}+\beta_{12}{exercise}_{it}+\beta_{13}{health\_anxiety}_{it}+\beta_{14}{loneliness}_{it}\\ & +\beta_{15}{myopic\_view}_{it}+\beta_{16}{smartphone}_{it}+u_{i}+\varepsilon_{it} \end{array}$$
where $u_{i}$ represents the unobserved heterogeneous factor and $\varepsilon_{it}$ is the error term. In addition to the full-sample analyses, we created a subsample based on gender and remote working status. We used STATA 18 statistical software to conduct the regression analysis.

## 4. Empirical Results

To test our hypothesis, we ran random-effects panel regressions for five emotional states: neutral, happy, angry, relaxed, and sad. For each emotional state, we used two models: one with only work-related control variables and the other with work-related and personal-characteristic-related control variables. In both models, the treatment group that had access to real-time feedback on emotions was the main independent variable. Table 3 presents regression coefficients and standard errors of the full-sample regression results. Our results showed that the treatment group had a significant negative association with happy states and a significant positive association with angry and sad states. However, the treatment group was not significantly related to neutral and relaxed states. Among the control variables, worktime had a significantly positive association with angry states, breaktime had a significantly negative association with angry states, being male had a positive relationship with neutral states but a negative association with angry and sad states, age had a negative relationship with angry states but a positive association with relaxed states, being married was positively associated with angry and sad states, university degree was negatively related to sad states, living alone was negatively associated with neutral states but positively associated with angry states, and having children and exercising were positively associated with sad states. Among psychological and cognitive factors, loneliness was positively associated with happy and relaxed states but negatively associated with angry states. Finally, myopic view of the future was positively associated with sad states.

To check the consistency of our results, we created two subsamples based on sex and ran a similar regression. The regression results for the male and female subsamples are presented in Table 4 and Table 5, respectively. In the gender subsample analyses, we found that among males, the treatment group was negatively associated with happy states and positively associated with angry and sad emotional states. Among control variables, worktime had a positive association and breaktime had a negative association with angry states; age had positive associations with happy and sad states but negative associations with neutral and angry states; being married had a negative association with neutral states but a positive association with sad emotional states; university degree had a positive association with sad states; living alone had a negative association with neutral states but a positive association with sad states; having children had a negative association with neutral states but positive associations with relaxed and sad states; travel time had negative associations with angry and sad states; household assets had a positive association with neutral states but a negative association with sad states; exercise had positive associations with relaxed and sad states; loneliness had a positive association with happy states but negative associations with neutral, angry, and sad states; myopic view had a negative association with happy states but positive associations with relaxed and sad states; smartphone had a positive association with neutral states but a negative association with sad states; and finally, health anxiety had a negative association with sad emotional states.

Among females, the treatment group was negatively associated with happy and angry states and positively associated with neutral and relaxed states. Among the control variables, breaktime had a negative association with happy states but positive associations with relaxed and sad states; age had negative associations with neutral, angry, and happy states but positive associations with relaxed and sad states, being married had positive associations with happy and relaxed states but a negative association with angry states; university degree had negative associations with neutral, happy, and angry states but a positive association with relaxed states; living alone had negative associations with neutral and angry states but positive associations with happy and relaxed states; having children had positive associations with neutral, angry, and sad states but negative associations with happy and relaxed states; travel time had positive associations with relaxed and sad states but negative associations with happy and angry states; asset had positive associations with happy and angry states but negative associations with neutral and relaxed states; exercise had positive associations with all states except happy states; loneliness had positive associations with happy and relaxed states but negative associations with angry and sad states; myopic view of the future had positive associations with neutral and relaxed states but negative associations with happy and angry emotional states; and finally, smartphone had a positive association with relaxed states but negative associations with happy and angry emotional states.

Given that there is a possibility of multicollinearity among control variables, we conducted correlation and multicollinearity tests on the models. The results showed that the multicollinearity between variables was not significant, suggesting that the independent effects of feedback on emotional states were not biased. Our VIF (variance inflation factor) statistics for the independent variables were less than 10, indicating that multicollinearity was not significant in the models (the results are not shown here, to save space, but are available upon request).

## 5. Discussion

The use of biometric devices to track real-time emotional changes has become a popular subject in academia [1,2,41,42]. However, only a few studies have analyzed these measurements in association with work-related activities, productivity, and performance [1,2]. Assessing the role of emotions in productivity by using subjective measurements is common [28]. Nevertheless, the inherent limitations of these measurements cannot be ignored. A significant benefit of using objective real-time emotional measurements is that it provides immediate feedback to participants so that they can modify their emotions if needed [1,2]. However, to date, there has been no study that explores the correlation between providing real-time emotional feedback and the subsequent modification of participants’ emotions in the workplace. As negative emotions are detrimental to well-being, productivity, and performance, emotional feedback could be a tool for mitigating stress and improving productivity [1]. Thus, this study contributes to the literature on bio-feedback and emotional regulation and its impact on employee well-being and productivity.

Our study provides evidence that receiving feedback on emotions has a significantly negative association with happy emotional states and a significantly positive association with angry and sad emotional states. Our results show that participants in the treatment group experienced considerably lower happy emotions than the control group and could not engage with positive emotions, even after receiving feedback. This failure to continue with favorable emotions is somewhat contradictory to previous findings [22,23]. Crucially, the negative association between happiness and the treatment group had productivity consequences, which is in line with the findings of previous studies on emotions and work [1].

Our study further found that participants in the treatment groups experienced considerably more time in sad and angry states compared with the control group, and that these emotions continued even after receiving feedback. Although it is expected that people modify their behavior when experiencing negative emotions [13,24], it appears that emotions such as sadness and anger can also trigger people to be aggressive or continue to be angry [44]. Similar to previous findings, this has productivity consequences, as continuing to experience sad and angry emotions could be detrimental to productivity [1,45]. Notably, our study did not find a significant association between the treatment group and relaxed or neutral emotions. Neutral and relaxed emotions exist in the less extreme domain of the emotion spectrum outlined by the Russel circumplex model [10,43]. Therefore, participants may not be as perplexed when experiencing neutral and relaxed emotions and may not care about modifying their emotions.

When interpreting the results of our study, it should be noted that feedback indicated that the treatment group was given access to the URL where they could see their real-time emotional conditions via a smartphone. Our study did not ensure that the treatment group members accessed the URL regularly or were taught coping strategies when faced with a particular type of emotion. Our study was designed to observe the reactions of the treatment group and their coping strategies. The negative association with the happy emotional state signals that the treatment group’s mood changed from happiness to something else. The positive association with sad and angry states signals that the provision of feedback could not prevent participants from continuing with negative emotions.

There are two possible explanations for this behavior. First, if the treatment group did not see the feedback and adopted any coping strategy to continue with the happy emotional state, the change indicated that the happy states did not continue for long among the insurance company employees. This pattern could be associated with worktime pressure and other similar conditions. Second, if the treatment group accessed the URL and observed their emotional conditions, this change from happy emotions to another indicated a lack of control over their emotions and the inability to adopt an effective coping strategy. In either case, the change in happy emotions due to feedback needs to be further studied, and the concerned organization must educate its employees on how to control and cope with emotions. Similar explanations are valid for the association between the treatment group and sad or angry emotional states. These explanations are consistent with Millgram et al. [46], who found individual differences in the degree of motivation to experience emotions and the subsequent degree of self-control or changes in emotions.

The subsample analysis reveals that the overall findings align with male employees who were unable to sustain happy emotions and instead transitioned into feelings of anger and sadness, even after receiving feedback. In contrast, female employees did not maintain their happy states but were able to persist in relaxed and neutral emotional states while reverting to anger. These findings, which indicate that female employees altered their behavior in response to emotional states, are in agreement with theories of behavioral modification [14,22,23].

An angry state, which was the most common observable emotion among the participants, has multiple covariates including longer worktime, shorter breaktime, being female, younger, married, and living alone (not loneliness). These findings are mostly consistent with stress theory [47] and are supported by previous studies [1,2].

Our study had some limitations. First, a biometric device, which is a mechanical object, may occasionally fail to track emotions. Sometimes, the device showed no emotionality, which we considered neutral. Second, the treatment was randomly assigned and allowed access to the observation of real-time emotions. However, our study did not determine whether the participants had access to the results. Third, the sample size of this study is not large enough to ensure a higher degree of power and precision of results. Moreover, because this study was carried out among the employees of an insurance company, its generalizability should be interpreted with caution. Nevertheless, our study provides additional evidence on the relationship between feedback on real-time emotions in the workplace and subsequent emotions and behavior. Future studies should include representative samples from various industries to better track whether the treatment group has regular access to the results. Moreover, workers need to be acquainted with emotion-coping strategies so that they can effectively shift their moods toward favorable emotions.

## 6. Conclusions

In examining the influence of real-time emotional feedback on self-modification and mental health, this study has contributed to the evolving discourse surrounding the correlation between emotional monitoring and workplace behavior. Despite the increasing interest in real-time emotional measurement through biometric devices and its link to productivity and safety, few studies have explored how this feedback impacts subsequent behavior within workplace settings. Our research addresses this gap by conducting a panel study involving 30 employees from an insurance company, analyzing their emotions over a 10-day period using attached biometric devices, and comparing the results between those who accessed their emotional feedback (treatment group) and those who did not (control group).

The findings shed light on a nuanced relationship between real-time emotional access and emotional states. Surprisingly, access to real-time emotions showed a negative association with happiness and a positive correlation with anger and sadness. Contrary to expectations, participants did not maintain happiness or redirect negative emotions toward more positive states despite having access to emotional feedback. This inability to effectively use feedback for self-modification may pose challenges to productivity and safety within the workplace. Given the absence of incentivization or coping strategies provided during this study, further research is warranted to investigate how workers engage with and utilize feedback systems for self-modification when properly motivated or trained.

The implications of our study are twofold for organizations, providing insights into both productivity enhancement and mental wellness strategies. While the feedback system was intended to empower users with emotional insights in real-time for behavioral modification, our findings indicate a potential gap between access and effective utilization. This underscores the critical need for organizations to offer comprehensive training programs that educate employees about using real-time emotional feedback for behavioral adaptation. By equipping individuals with the tools to adjust behaviors based on past emotional signals, organizations can foster a workplace culture conducive to improved mental wellness and enhanced productivity.

Furthermore, for individuals who face challenges in behavioral modification, exploring additional approaches such as stimulation or medication could offer alternative avenues of improvement. As organizations navigate the implementation of real-time emotional feedback systems, it becomes imperative to not only provide access but also empower employees with the skills and resources necessary to harness this information effectively for their well-being and productivity.

## Figures and Tables

**Table 1 behavsci-14-00169-t001:** Variable definitions.

Variable	Definition
p_neutral	Percentage of working hours in which workers do not show any particular emotion
p_happy	Percentage of working hours in which workers remain in a happy state
p_angry	Percentage of working hours in which workers remain in an angry state
p_relaxed	Percentage of working hours in which workers remain in a relaxed state
p_sad	Percentage of working hours in which workers remain in a sad state
Treatment	Binary variable: equal to 1 if the respondents are in the treatment group
Worktime	Continuous variable: working time per day (/m)
break_walk_exercise	Binary variable: equal to 1 if the respondents walked/exercised when they took a break
male	Equal to 1 if the respondents are male
age	Age of the respondents
Married	Equal to 1 if the respondents are married
university_degree	Binary variable: equal to 1 if the respondents have a bachelor degree
living_alone	Binary variable: equal to 1 if the respondents are living alone
Children	Equal to 1 if the respondents have at least one child
travel_time	Continuous variable: commuting time (one-way) of the respondents
hasset	The respondents’ annual household financial assets
log_of_hasset	Natural log of the respondents’ annual household financial assets
Exercise	Equal to 1 if the respondents exercise at least twice a week
health_anxiety	Binary variable: equal to 1 if the following statement is true or partially true for the respondents: “I am anxious about my health” before the experiment
Loneliness	Binary variable: equal to 1 if the respondents frequently/occasionally felt “a lack of companionship”, “left out”, or “isolated from others” before the experiment
myopic_view	Binary variable: equal to 1 if the following statement is true or partially true for the respondents: “Since the future is uncertain, it is a waste to think about it” before the experiment
smartphone	Continuous variable: usual time of the respondents’ smartphone use before the experiment (/m)

**Table 2 behavsci-14-00169-t002:** Descriptive statistics.

Variables	Mean	SD	Min	Max	Observation
p_neutral	0.2129		0	0.8838	277
p_happy	0.2819		0	0.8775	277
p_angry	0.3590		0	0.8583	277
p_relaxed	0.1035		0	0.5813	277
p_sad	0.0427		0	0.3768	277
Treatment	0.5000		0	1	277
Worktime	630.0238	96.6332	180	930	294
travel_time	53.5204	19.1617	30	105	294
break_walk_exercise	0.0374	0.1901	0	1	294
Male	0.5667		0	1	277
Age	37.5667	9.9088	23	59	277
Married	0.5667		0	1	277
University_degree	0.8667		0	1	277
living_alone	0.3667		0	1	277
Children	0.1667		0	1	277
Hasset	129,000,000	18,600,000	2,500,000	75,000,000	277
log_hasset	15.7964	0.9799	14.7318	18.133	277
Exercise	0.1667		0	1	277
health_anxiety	0.3667		0	1	277
Loneliness	0.6000		0	1	277
myopic_view	0.4667		0	1	277
Smartphone	263.6667	207.9485	2	840	277

**Table 3 behavsci-14-00169-t003:** Estimation results.

Variables	Neutral		Happy		Angry		Relaxed		Sad	
treatment	0.0509	0.0425	−0.0998 *	−0.1370 *	0.0736	0.133 ***	−0.0385	−0.0507	0.0126	0.0134 **
	−0.0482	(0.0430)	(0.0571)	(0.0784)	(0.0636)	(0.0495)	(0.0434)	(0.0314)	(0.0087)	(0.0052)
worktime	−0.0001	−0.0002	−0.0000	−0.0000	0.0002 **	0.0002 ***	0.0000	−0.0000	−0.0023	−0.0000
	(0.0001)	(0.0001)	(0.0000)	(0.0000)	(0.0001)	(0.0001)	(0.0000)	(0.0000)	(0.0000)	(0.0000)
break_walk_exercise	0.0587	0.0721	−0.0325	−0.0365	−0.0975 **	−0.1080 **	0.0550	0.0661	−0.0098	−0.0056
	(0.0772)	(0.0725)	(0.0461)	(0.0481)	(0.0463)	(0.0487)	(0.0443)	(0.0459)	(0.0104)	(0.0116)
male		0.175 ***		−0.0035		−0.1050 *		−0.0463		−0.019 ***
		(0.0476)		(0.0818)		(0.0618)		(0.0416)		(0.0052)
age		−0.00175		0.0040		−0.0117 ***		0.0085 ***		0.0007
		(0.0032)		(0.0042)		(0.0028)		(0.0021)		(0.0004)
married		−0.131		−0.0475		0.1660 ***		−0.0063		0.0196 **
		(0.0807)		(0.0911)		(0.0479)		(0.0325)		(0.0078)
university_degree		−0.0657		0.1320		−0.0446		0.0009		−0.0223 *
		(0.0806)		(0.1060)		(0.0945)		(0.0972)		(0.0123)
living_alone		−0.130 **		−0.0549		0.175 ***		0.0016		0.00743
		(0.0661)		(0.0879)		(0.0500)		(0.0383)		(0.0065)
children		−0.0727		−0.0153		0.0954		−0.0295		0.0215 **
		(0.0724)		(0.118)		(0.0743)		(0.0488)		(0.0086)
travel_time		0.00116		−0.0009		−0.0000		−0.0003		0.0001
		(0.0015)		(0.00193)		(0.0014)		(0.0010)		(0.0001)
log_hasset		0.0082		−0.00789		0.0178		−0.0190		0.0007
		(0.0192)		(0.0268)		(0.0215)		(0.0187)		(0.0030)
exercise		0.0495		−0.117		0.0671		−0.0313		0.0305 ***
		(0.0581)		(0.0789)		(0.0646)		(0.0601)		(0.0070)
health_anxiety		−0.0852		0.0539		0.0105		0.0216		−0.0005
		(0.0538)		(0.0860)		(0.0655)		(0.0364)		(0.0081)
loneliness_ucla		−0.0595		0.120 **		−0.0967 **		0.0535 *		−0.017 ***
		(0.0396)		(0.0583)		(0.0422)		(0.0303)		(0.0051)
myopic_view		−0.0001		−0.0327		−0.0129		0.0172		0.0263 ***
		(0.0613)		(0.0719)		(0.0499)		(0.0450)		(0.0078)
Smartphone		−0.0000		−0.0000		0.0000		−0.0000		−0.0000
		(0.0002)		(0.0002)		(0.0000)		(0.0000)		(0.0000)
Constant	0.296 ***	0.360	0.373 ***	0.305	0.158 **	0.225	0.115 ***	0.120	0.0520 ***	0.0009
	(0.110)	(0.327)	(0.0766)	(0.455)	(0.0676)	(0.322)	(0.0333)	(0.269)	(0.0169)	(0.0378)
Observations	277	277	277	277	277	277	277	277	277	277

Note: Standard errors are in parentheses. *** *p* < 0.01, ** *p* < 0.05, * *p* < 0.1.

**Table 4 behavsci-14-00169-t004:** Estimation results for male subsample.

Variables	Neutral		Happy		Angry		Relaxed		Sad	
Treatment	−0.0184	0.0350	−0.122	−0.232 ***	0.126	0.191 ***	−0.00414	−0.0236	0.0237 **	0.0204 ***
	(0.0641)	(0.0430)	(0.109)	(0.0633)	(0.0921)	(0.0460)	(0.0358)	(0.0219)	(0.0104)	(0.00356)
Worktime	−0.00034 *	−0.0002	−0.0000	−0.00012	0.00038 **	0.00047 **	−0.0000	−0.0000	−0.0000	−0.0000
	(0.0002)	(0.0002)	(0.0001)	(0.0001)	(0.0001)	(0.0001)	(0.0000)	(0.0000)	(0.0000)	(0.0000)
break_walk_exercise	0.123	0.126	−0.0220	−0.0202	−0.125 *	−0.140 **	0.0394	0.0415	−0.0134	−0.0156
	(0.0854)	(0.0966)	(0.0762)	(0.0798)	(0.0692)	(0.0713)	(0.0647)	(0.0668)	(0.0163)	(0.0187)
Age		−0.00854 **		0.0106 **		−0.0128 ***		0.00830 ***		0.00192 ***
		(0.0033)		(0.0050)		(0.00314)		(0.0007)		(0.0001)
Married		−0.155 ***		0.00142		0.0708		0.0364		0.0368 ***
		(0.0376)		(0.0630)		(0.0630)		(0.0257)		(0.0083)
university_degree		−0.496		0.183		-		-		0.341***
		(0.574)		(0.779)		-		-		(0.0544)
living_alone		−0.174 **		−0.113		0.189		0.0421		0.0413 ***
		(0.0828)		(0.147)		(0.119)		(0.0415)		(0.0076)
Children		−0.186 ***		0.103		−0.0151		0.0613 ***		0.0276 ***
		(0.0549)		(0.100)		(0.0751)		(0.0219)		(0.0062)
travel_time		0.00219		0.0049		−0.0054 **		−0.0004		−0.0014 ***
		(0.0021)		(0.0030)		(0.0026)		(0.00117)		(0.0001)
log_hasset		0.0808 **		−0.0236		−0.0075		−0.0249		−0.0209 ***
		(0.0326)		(0.0480)		(0.0425)		(0.0190)		(0.0025)
Exercise		−0.0780		−0.102		0.0269		0.0867 ***		0.0630 ***
		(0.0492)		(0.0902)		(0.0841)		(0.0271)		(0.00621)
loneliness_ucla		−0.0998 *		0.311 ***		−0.209 ***		0.0357		−0.0460 ***
		(0.0517)		(0.0762)		(0.0500)		(0.0222)		(0.0029)
myopic_view		−0.0595		−0.205 ***		0.111		0.0780 **		0.0832 ***
		(0.0548)		(0.0771)		(0.0948)		(0.0352)		(0.0050)
Smartphone		0.000227 *		−0.000110		0.0000		−0.0000		−0.0000 ***
		(0.0548)		(0.0771)		(0.0948)		(0.0352)		(0.0050)
health_anxiety		0.0630		0.124		−0.0727		−0.0495		−0.0548 ***
		(0.0735)		(0.133)		(0.120)		(0.0437)		(0.00624)
Constant	0.485 ***	−	0.409 ***	−	0.0372	0.762	0.0901 ***	0.139	0.0311	0.485 ***
	(0.152)		(0.138)		(0.125)	(0.719)	(0.0332)	(0.341)	(0.0215)	(0.152)
Observations	154	154	154	154	154	154	154	154	154	154

Note: Standard errors are in parentheses. *** *p* < 0.01, ** *p* < 0.05, * *p* < 0.1.

**Table 5 behavsci-14-00169-t005:** Estimation results for female subsample.

Variables	Neutral		Happy		Angry		Relaxed		Sad	
Treatment	0.0654	0.201 ***	−0.0804	−0.0481 **	0.0413	−0.5150 ***	−0.0384	0.3630 ***	0.0089	−0.0003
	(0.0963)	(0.0371)	(0.0585)	(0.0202)	(0.122)	(0.0213)	(0.0801)	(0.0140)	(0.0118)	(0.0074)
Worktime	−0.00012	−0.0001	−0.0000	−0.0000	0.0001	0.0001	0.0000	0.0000	−0.0000	−0.0000
	(0.0002)	(0.0002)	(0.0001)	(0.0001)	(0.0001)	(0.0001)	(0.0000)	(0.0000)	(0.0000)	(0.0000)
break_walk_exercise	−0.0106	0.0103	−0.040 ***	−0.0638 ***	−0.041 ***	−0.0148	0.0740 ***	0.0585 ***	−0.006 **	0.0098 **
	(0.0184)	(0.0244)	(0.0106)	(0.0133)	(0.0141)	(0.0140)	(0.0095)	(0.0091)	(0.0030)	(0.0048)
Age		−0.0030 **		−0.0140 ***		−0.0290 ***		0.0454 ***		0.0016 ***
		(0.0013)		(0.0007)		(0.0007)		(0.0005)		(0.0002)
Married		−0.1210		0.140 ***		−0.7200 ***		0.7020 ***		−0.0016
		(0.0782)		(0.0426)		(0.0447)		(0.0295)		(0.0156)
university_degree		−0.149 **		−0.2960 ***		−1.1620 ***		1.6040 ***		0.0036
		(0.0701)		(0.0382)		(0.0401)		(0.0264)		(0.0140)
living_alone		−0.2420 ***		0.1090 ***		−0.3850 ***		0.5070 ***		0.0115
		(0.0521)		(0.0284)		(0.0298)		(0.0196)		(0.0104)
Children		0.3140 ***		−0.3450 ***		0.3350 ***		−0.3260 ***		0.0215 ***
		(0.0267)		(0.0145)		(0.0153)		(0.0100)		(0.0053)
travel_time		0.0010		−0.0090 ***		−0.0210 ***		0.0283 ***		0.0011 ***
		(0.0008)		(0.0004)		(0.0005)		(0.0003)		(0.0001)
log_hasset		−0.0930 ***		0.1130 ***		0.3330 ***		−0.3560 ***		0.0029
		(0.0221)		(0.0120)		(0.0126)		(0.0083)		(0.0044)
Exercise		0.1100 ***		−0.3840 ***		0.1850 ***		0.0296 ***		0.0589 ***
		(0.0268)		(0.0146)		(0.0153)		(0.0101)		(0.0053)
loneliness_ucla		0.0494		0.2140 ***		−0.5480 ***		0.3150 ***		−0.0299 ***
		(0.0433)		(0.0236)		(0.0248)		(0.0163)		(0.0086)
myopic_view		0.0960 *		−0.331 ***		−0.861 ***		1.0510 ***		0.0458 ***
		(0.0564)		(0.0307)		(0.0323)		(0.0213)		(0.0113)
Smartphone		0.000165		−0.0003 ***		−0.003 ***		0.0028 ***		0.0000
		(0.000190)		(0.000104)		(0.000109)		(0.0000)		(0.0000)
Constant	0.212	1.801 ***	0.347 ***	−0.156 ***	0.264 ***	0.00845	0.108 **	−0.535 ***	0.0568 **	−0.119 ***
	(0.141)	(0.0861)	(0.109)	(0.0469)	(0.0534)	(0.0493)	(0.0517)	(0.0324)	(0.0256)	(0.0172)
Observations	123	123	123	123	123	123	123	123	123	123

Note: Standard errors are in parentheses. *** *p* < 0.01, ** *p* < 0.05, * *p* < 0.1.

## Data Availability

Data are available upon request.

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
