# Peer review of "Could Having Access to Real-Time Data on Your Emotions Influence Subsequent Behavior? Evidence from a Randomized Controlled Trial of Japanese Office Workers"

_behavsci, 2024, doi:10.3390/bs14030169_

Round 1

Reviewer 1 Report (Previous Reviewer 2)

Comments and Suggestions for Authors

Thank you for providing me with the opportunity to, once again, review your submitted manuscript titled “Could having access to real-time data on your emotions help improve mental health? Evidence from a Randomized Controlled Trial of Japanese Office Workers”, now in its new reviewed version (3rd overall). Below, I provide a point-by point review of some the   concerns I still have and which will help the authors refine the current version of the paper.

1. The introduction section has benefit from a substacial improvement when compared with the last version of the manuscript. It now frames and explains the rational as well as the structure of the manuscript in a much more robust manner .  

2. The literature review, although it could have been yet better worked, does now facilitate the understanding of the knowledge gap identified by the authors. However, it is still not clearly stated the research preposition that the paper is addressing. Once more, I strongly recommend adding a paragraph at the end of this section, framing, deriving and developing the research hypothesis/question/preposition.

3. The main findings and conclusions are sustained by the results of the research and consistent with the evidence. Furthermore, limitations are fairly considered and presented. It would have been more useful to have gathered a larger and more representative sample of respondents, but even so, the added value of the paper cannot be neglected.

Comments on the Quality of English Language

The use of English language is, in general, good.

Author Response

Reviewer 2 Report (Previous Reviewer 4)

Comments and Suggestions for Authors

Thank you for the opportunity to review the revised version of the manuscript “Could having access to real-time data on your emotions help improve mental health? Evidence from a Randomized Controlled Trial of Japanese Office Workers” submitted to Behavioral Sciences (Manuscript ID: behavsci-2763608). Unfortunately, the authors haven’t implemented my suggestions and haven’t explained why. 

My critical concerns about the paper are related to the following issues:

·       The research problem should be more clearly described in the introduction. It is not entirely clear why the issues raised in the manuscript are important. Just because something hasn't been researched yet doesn't mean it's worth investigating.

·       The literature review needs to be supplemented with new literature and reorganized. This section contains some elements related to the description of the research gap, which I would suggest moving to the introduction.

·       I propose to highlight the proposed hypothesis more clearly. The literature review should be reorganized to show step-by-step the development and justification for the mentioned hypothesis.

·       The practical implications section is vague. I recommend the authors be more specific with what these findings can be used for. Given many of the effects have been found previously, but the current manuscript just does it all in one place, really strengthening the practical recommendations would bolster the overall contribution of the manuscript.

·       The manuscript requires minor linguistic corrections, e.g. "However, to date, no studies HAVES (?) investigated the relationship between the provision of real-time emotional feedback and the subsequent modification of participants' emotions in the workplace."

Comments on the Quality of English Language

Thank you for the opportunity to review the revised version of the manuscript “Could having access to real-time data on your emotions help improve mental health? Evidence from a Randomized Controlled Trial of Japanese Office Workers” submitted to Behavioral Sciences (Manuscript ID: behavsci-2763608). Unfortunately, the authors haven’t implemented my suggestions and haven’t explained why. 

My critical concerns about the paper are related to the following issues:

·       The research problem should be more clearly described in the introduction. It is not entirely clear why the issues raised in the manuscript are important. Just because something hasn't been researched yet doesn't mean it's worth investigating.

·       The literature review needs to be supplemented with new literature and reorganized. This section contains some elements related to the description of the research gap, which I would suggest moving to the introduction.

·       I propose to highlight the proposed hypothesis more clearly. The literature review should be reorganized to show step-by-step the development and justification for the mentioned hypothesis.

·       The practical implications section is vague. I recommend the authors be more specific with what these findings can be used for. Given many of the effects have been found previously, but the current manuscript just does it all in one place, really strengthening the practical recommendations would bolster the overall contribution of the manuscript.

·       The manuscript requires minor linguistic corrections, e.g. "However, to date, no studies HAVES (?) investigated the relationship between the provision of real-time emotional feedback and the subsequent modification of participants' emotions in the workplace."

Author Response

Reviewer 3 Report (Previous Reviewer 5)

Comments and Suggestions for Authors

dear author.

Thank you for your corrections. Despite your great work, I still see the following major points for improvement in the drafting of your report which affect the quality of the report.

The background is unstructured, excessively flat, and needs to be completely modified in its structure and content. The methodological section is confusing, not describing such basic processes in the research as the determination of the number of members of the sample, the sampling process, the description of the measurement instruments and their psychometric properties.

The results are confusing and unclear, so that they cannot be interpreted accurately. 

The conclusions are excessively long, and do not respond to the specific results of the research; they incorporate assessments that do not come from the research itself, but from attributions made by the researchers. Conclusions should not incorporate these aspects, but are a faithful and affirmative reflection of the research objectives and/or hypotheses.

I hope these indications will serve as a corrective to this interesting research work. 

Regards

Author Response

This manuscript is a resubmission of an earlier submission. The following is a list of the peer review reports and author responses from that submission.

Round 1

Reviewer 1 Report

Comments and Suggestions for Authors

The paper has done significant revisions and meets the publication requirements. However, the shortcomings of the paper should also be supplemented, which is determined by the methodology and research subjects of this study. For example, the number of males and females is relatively small, neither reaching 30, which may affect the final outcome.

Comments on the Quality of English Language

very well

Reviewer 2 Report

Comments and Suggestions for Authors

Thank you for providing me with the opportunity to, once more, read and review your submitted manuscript titled “Could having access to real-time data on your emotions help improve mental health? Evidence from a Randomized Controlled Trial of Japanese Office Workers”, now in its second and reviewd version. Below, I provide a point-by point review of some of my  concerns which, I hope, will help the authors refine the current version of the paper.

1. The introduction section has benefit from a substacial improvement when compared with the first version. It now frames and explains the rational as well as the structure of the manuscript in a much more robust manner .  

2. The literature review, although it could have been still better worked, does facilitate the understanding of the knowledge gap identified by the authors. However, the hypothesis/research question development or at least the research preposition needs to be better addressed and explained. It is not clearly stated the research preposition that the paper is addressing. I strongly recommend adding a paragraph at the end of this section, framing, deriving and developing the research hypothesis/question/preposition.

3. The research method is adequate, although the paper still lacks some important details that would have been useful in better understanding the employed methodology. Moreover, when reporting and explaining table 2, (lines 267 to 279) the authors report percentages and standard deviations? I do not understand this. This needs to be reviewed and edited.

4. The main findings and conclusions are sustained by the results of the research and consistent with the evidence. Furthermore, limitations are fairly considered and presented. It would have been more useful to have gathered a larger and more representative sample of respondents, but even so, the added value of the paper cannot be neglected.

5. Both limitations and conclusion have been somehow improved.

Comments on the Quality of English Language

The use of English language is, in general, good. However the paper can benefit from a further overall proofreading.

Reviewer 3 Report

Comments and Suggestions for Authors

The authors did not addres my main concerns adequately, but addressed only some minor stylistycal ones.

The methodology section seems very poor, and it was not reconsidered adequately.

Statistical errors are obvious: For example, the authors stated that they have 30 observations on loneliness (Table 2), whereas in Table 3 they indicated that there are 277 observations on loneliness. The statistical analysis was conducted incorrectly.

I do not trust your analyses due to obvious errors.

Reviewer 4 Report

Comments and Suggestions for Authors

I had the pleasure of reviewing the manuscript: “Could having access to real-time data on your emotions help improve mental health? Evidence from a Randomized Controlled Trial of Japanese Office Workers.” submitted to the Behavioral Sciences (Manuscript ID: behavsci-2650075). Although I have read the entire manuscript with interest, I have some critical concerns about the paper. I hope the author(s) will find my comments useful in the development of their work.

1/ The research problem should be more clearly described in the introduction. It is not entirely clear why the issues raised in the manuscript are important. Just because something hasn't been researched yet doesn't mean it's worth investigating.

2/ The literature review needs to be supplemented with new literature and reorganized. This section contains some elements related to the description of the research gap, which I would suggest moving to the introduction.

3/ I propose to highlight the proposed hypothesis more clearly. The literature review should be reorganized to show step-by-step the development and justification for the mentioned hypothesis.

4/ The practical implications section is vague. I recommend the authors be more specific with what these findings can be used for. Given many of the effects have been found previously, but the current manuscript just does it all in one place, really strengthening the practical recommendations would bolster the overall contribution of the manuscript.

5/ The manuscript requires minor linguistic corrections, e.g. "However, to date, no studies HAVES (?) investigated the relationship between the provision of real-time emotional feedback and the subsequent modification of participants' emotions in the workplace."

Reviewer 5 Report

Comments and Suggestions for Authors

Dear author.

Thanks for your papper. I enjoyed reading it, I found it an interesting topic. However, I would like to make a series of clarifications in this regard.

a) the writing of the research report is disorganized into a large number of sections and aspects, collecting information related to the methodology of the work in the foundation; data related to the research, in the methodology section, etc. That is why it must be reviewed in its entirety and refined according to international standards for editing scientific texts.

b) the research protocol is not clearly described; The randomization process of the control and intervention groups, the purpose of said intervention, and what it consists of are not clearly clear. Therefore, it must be redrafted in a clear manner.

c) The sample selection process, the sample size selection process, the choice of the company and no other, the sampling process, selection of participants, inclusion and exclusion criteria, are not described.

d) regarding the medication of emotions, the recording of generic biometric data such as heart rate, physical activity, respiratory rate, etc., taken using an activity bracelet, seems insufficient evidence to associate such basic emotions with such as joy, sadness, anger, etc... recorded in your research; At least a close relationship is not established despite the evidence provided. The analysis of emotions has been carried out in previous studies with much more specific objective determinations such as electroencealography, and subjective response analysis, or interviews with emotion managers. like psychologists, .... with the intention of making subjects aware of their emotions, but guided by an expert. I am not able to understand how the heart rate changes. can help identify the emotion as anger or joy.

Therefore,  you must rethink the research report with the intention of clarifying the purpose, the procedure followed and the results obtained.

Thanks for your contributions.

Kind regards.